# Advances in Flow of Water Through Variably Saturated Soils: A Review of Model Approaches and Experimental Investigations with Use of Sensors

**DOI:** 10.3390/s25227027

**Published:** 2025-11-17

**Authors:** Peter Uloho Osame, Ebikapaye Peretomode, Haval Kukha Hawez

**Affiliations:** 1School of Computing, Engineering, and Technology (SoCET), Robert Gordon University, Aberdeen AB10 7GJ, UK; e.peretomode@rgu.ac.uk; 2Department of Petroleum Engineering, Faculty of Engineering, Koya University, Koya KOY45, Iraq; haval.hawez@koyauniversity.org

**Keywords:** characterisation, experimental investigation, hydraulic properties, model approaches, review, subsurface flow

## Abstract

The study of the flow of water through soils is important and has applications in many fields such as irrigation in agriculture, engineering, hydrogeology, and earth sciences. Many research efforts have been focused on different aspects of the subject of flow through soils. These include flow through the vadose zone where the flow is transient, saturated flow, soil water evaporation, Darcian or laminar flow, macroporous or differential flow, flow through homogeneous soils, and flow through heterogeneous soils. Although Darcy’s law is the most fundamental law governing soil water subsurface flow, it considers a linear relation between flow velocity and pressure gradient. Formulation of Darcy’s law is based on steady flow of incompressible liquid when the porous medium is isotropic, homogeneous, and saturated. However, these classical representations of water flow are not adequate when considering flow through natural soils, due to influences caused by the existence of macropores and spatial variability of soil properties. Despite researchers’ non-linear models which modify Darcy’s law, such as Richard’s equation for transient unsaturated flow of water in soils, determination of soil hydraulic properties also requires other techniques and measurement methods. This study focuses on model approaches and experimental investigations of water flow through the soil subsurface with instruments and sensors for determination of hydraulic properties and parameters for flow characterisation. It critically examines challenges and the accuracy of best practices and aims to present novel methods of experimental approach for potential solutions.

## 1. Introduction

Applications for studying how water moves through soil are significant in a variety of domains, including hydrogeology, engineering, geosciences, and agricultural irrigation. However, flow laws must be used in order to properly understand how water moves through soils, which makes measuring the hydraulic properties of the soil crucial. Since the amount of soil is made up of the soil matrix and vacant spaces that create pores, when the soil is saturated, water fills every pore [1]. Some of the pores in unsaturated soils, sometimes referred to as the vadose zone [2,3,4,5,6], fill with air, which lowers conductivity. Between the two extremes are gravitational water that freely drains to reach field capacity, accessible water that plants can absorb, and unavailable water that plants cannot use. The second of these terms refers to the water that remains after the soil has freely drained for 24 to 48 h and is also available for plant uptake.

Numerous researchers have taken different approaches to the hydrodynamic characterisation of soil subsurface flows [7], which take into account the spatial distributions of hydraulic properties and hydraulic states in the subsurface, including permeability, soil water content, soil matric potential, soil hydraulic conductivity, soil water diffusivity, and soil water retention characteristics. Examples of these models [8,9,10,11,12,13,14,15,16] include microstructure models, discrete macropore models, dual and multiple permeability models, geometry-based mobile–immobile transport models, first-order-type mobile–immobile transport models, and conventional single-domain and double-porosity models.

However, model-based research requires input parameters that control the soil’sability to retain chemicals and water. Many pedotransfer functions (PTFs) [17,18,19,20,21] have been established to forecast hydraulic parameters and the accompanying hydraulic functions; these are typically used to obtain input parameters such as textural soil qualities related to particle size distribution and porosity. However, PTFs are site-specific and should be used carefully because they frequently rely on statistical regression equations.

Direct measurement, such as field or in situ methods of measuring soil hydraulic characteristics or collecting and testing numerous samples in a laboratory, is typically used to obtain high-resolution characterisation. Even though direct measurement of hydraulic characteristics is becoming more advanced, most critical zone studies cannot afford these methods due to their high cost and need for specialised sensors and technician skills [22]. These methods can be laboratory-based or field-based, and include the use of tension disc infiltrometers, double ring infiltrometers, constant-head permeameters, or undisturbed soil core methods [23,24], and others. Because direct measurement is the only way to build the database used for the derivation and calibration of predicted hydraulic characteristics, it should be noted that no indirect method exists, even though it is well known that direct methods of measuring hydraulic properties are time-consuming and costly. Therefore, ongoing research on modern, precise, and effective direct measuring techniques is necessary for the development of indirect approaches.

This study reviews models of flow of water through variably saturated soils and experimental investigations of water flow through the soil using direct methods including laboratory soil column experiments and field experiments with the use of sensors to obtain hydraulic parameters.

## 2. Research Methodology

A literature review on water flow processes through saturated and unsaturated soils was carried out. While there is considerable research on water flow processes through variably saturated soils, the focus of this review is to study models of water flow through the soil subsurface and review experimental investigations on soil hydraulic properties with instruments and sensors. Materials for the literature review were obtained by searching keywords incorporating soil hydraulic properties such as permeability, soil water content, soil matric potential, soil water characteristics curve, soil hydraulic conductivity, and soil water diffusivity. Its scope also included hydrodynamic characterisation, models of flow through soil, soil subsurface flow, and soil column experiments with the use of sensors. The very rich Robert Gordon University LibGuides, which embeds engineering search tools such as Web of Science, Scopus, Science Direct, etc., were engaged, in addition to Google search machine.

This review is divided into the following three sections:
Water flow processes through variably saturated soil reviews—this section considers reviews on water flow processes through saturated and unsaturated soils for the purpose of the determination of the soil subsurface flow characteristics.Soil water flow models—this part discusses model approaches developed for different flow conditions.Experimental investigation of flow through variably saturated soils—this discussion is divided into in situ or field experiments and laboratory investigations of the flow of water through soil columns with the use of sensors.

After the literature review is presented, each experimental investigation is then individually analysed for simplicity, challenges of use, and accuracy of results. Following this, the future of use of sensors for soil water flow experiments is discussed. The structure of the review paper is shown in Figure 1

## 3. Review of Model Approaches

### 3.1. Reviews

In the application of flow laws for flow of water through soils, Darcy’s law is fundamental [25,26,27]; it considers a linear relation between flow velocity and pressure gradient in a packed sand bed. Darcy’s law is represented as follows [27,28]:
(1)$$Q=-KA\frac{dh}{dx}$$
where Q is total discharge in m^3^/s, K is hydraulic conductivity in m/s, A is cross sectional area in m^2^, and dh/dx is the hydraulic gradient, which is dimensionless. The negative sign is because the fluid flows down (negative) the hydraulic gradient from higher values to lower values. This is also written in the following form:
(2)$$Q=\frac{-KA\left(P_{b}-P_{a}\right)}{\mu L}$$
where (P_b_ − P_a_) is the pressure drop in Nm^−2^, µ is the fluid dynamic viscosity in Nsm^−2^, and L is the depth.

Formulation of Darcy’s law is based on steady, uniform flow of incompressible liquid where the porous medium is isotropic, homogeneous, and saturated. However, the natural soil environment can be heterogeneous in structure [22,29,30,31]. Water flow processes and solute transport in natural soils are significantly dominated by the occurrence of structural elements and macropores, which could be termed as micro-heterogeneous, and spatial variability of soil properties, which could be termed as macro-heterogeneous. The concept of homogeneity and heterogeneity of soils is dependent on whether the permeability is constant from one point to the other over the medium or varies from point to point. The flow can also be in unsaturated soil [32] and the flow can be preferential flow or macropore flow or non-equilibrium flow [16,33,34,35]. Heterogeneity is attributed to critical zones, whether in the vertical or lateral directions, because the interactions between the vegetation, climate, and bedrocks are complex. These conditions invalidate the application of the Darcy’s equation to the flow processes in its lone form. Hence, the most used model for soil water dynamics is represented by the Richards equation, which includes Darcy’s law extended to unsaturation with mass conservation [16,36,37]. The solution of this equation in both its analytical and numerical form is complex because of the nonlinear relationship that links soil water to soil hydraulic conductivity and soil matric potential.

A review of models for finger flow studies with an emphasis on the vadose zone was given by [38]. He identified eight theoretical expressions for the finger radius (Rf) using linear stability analysis (LSA), and he further divided them into three groups based on the ratio of the infiltration rate to the soil hydraulic conductivity in the finger. In Table 1, the expressions are displayed. The finger radius is summarised in theoretical expressions in Table 1, below. The subscripts f and i stand for the value in the finger and the initial value of the subscripted variable, respectively. Rf is the finger radius (m), g is the acceleration due to gravity [LT-2], η is a soil characteristic parameter, ρ is the fluid density [ML-3], θ is the soil volumetric water content, and σ is the bulk water–air surface tension MT-2. In his review of finger growth [39,40,41], ref. [38] states that the effects of the initial water content on finger development are unknown. It is impossible to say that finger size predictions are correct, despite the large number of papers from extensive study on fingered flow models. This is due to the fact that the large degree of heterogeneity found in natural soils influences the expressions for homogenous profiles.

With an emphasis on the vadose zone, ref. [52] examined a wide range of modelling techniques for preferential and non-equilibrium flow and transport. Depending on whether the model uses the Richard’s equation or kinematic wave equation, such as in [53], for flow in the soil structure with the assumption of immobile water in the soil matrix, or a combination of the Richard’s equation with composite equation of the hydraulic properties of the soil matrix and structure [54,55,56,57], they characterise the models as simple or more complex. Ref. [52] describes preferential flow in structured media using various dual-porosity, dual-permeability, multi-porosity, and/or multi-permeability models [58,59]; the dual-porosity and dual-permeability models make the assumption that a porous medium is made up of two interaction regions: the rock matrix, also known as a micropore, and the fracture system, also known as a macropore. The dual-permeability models assume water movement in both the matrix and the macropore or fracture system, whereas the dual-porosity model assumes flow in the macropore with stagnant water in the matrix. In contrast to models for single pore regions, ref. [52] states that the dual-porosity and dual-permeability models have the drawback of requiring an excessive number of input parameters [59] to characterise both pore systems. However, it is unclear how to obtain these parameters, either directly or through estimation [60,61]. The research provides an explanation of why models that describe preferred and/or non-equilibrium flow require intercode comparison, which led to the creation of HYDRIUS-ID and HYDRIUS-2D.

A review of model approaches for explaining preferred flow in structured soils was given by [62]. In [63], the soil matrix is classified as single-grained, massive, and aggregated. The research also distinguishes between uniform and nonuniform flow through soils [64,65]. Wetting fronts in uniform flow are stable and descend in a homogeneous front, however in nonuniform flow, the soil’s wetting profile is uneven. Uneven flow patterns are observed in nonuniform flow, where water flows across the soil profile more quickly in some places than others. This nonuniform flow pattern, also referred to as preferential flow, can take the following forms: finger-like flow, in which a portion of the soil is bypassed; funnel flow, in which less permeable zones are bypassed; unstable flow for coarse-textured soils; and macropore flow for root channels, cracks, fissures, and earthworm burrows [66]. Based on the convection–dispersion equation for solute transport and the conventional Richards equation for unsaturated and variably saturated flow, the author describes the standard single-domain models [65]. Additionally, he takes into account the discrete macropore models [67], dual- and multiple-permeability models [57,58,68], and double-porosity models [69,70,71,72,73]. He reviews deterministic methods for explaining preferential flow and transport in structured soil, with a focus on the two-domain approach. He asks whether capillary forces or gravity and viscous forces still affect water in preferential flow paths, and whether the domain concept, which assumes a porous continuum, applies to preferential flow paths as well as the soil matrix. He asserts that despite advancements, conceptual issues identifying the preferential flow domain remain unresolved and that methods for independent parameter determination and measurement procedures are still required.

With an emphasis on agricultural soils rather than rock cracks, ref. [74] reviews the effects, governing variables, and principles influencing non-equilibrium water flow and solute transport in soil macropores. That study argues that the representative elementary volume notion is invalidated by macropore flow, which is an occurrence of heterogeneous structures that produce non-uniform water pressure or solute concentration, or both, during vertical flow and conveyance. It poses a question regarding the pore size [75,76] that is adequate for non-equilibrium water flow and solute transport, and it posits that pores of equivalent cylindrical diameter that are greater than about 0.3–0.5 mm can be categorized as macropores [77,78,79,80]. This is equivalent to water-entry pressures of −10 to −6 cm of water in the Laplace equation.

The network of macropores is affected by biological activities such as root channels and earthworm burrows [62,81,82,83] and chemical activities such as leaching and the deposition of elements such as phosphorus [84,85], nitrates [86], and trace elements [87,88] which can affect the clay content [89,90]. According to [91,92,93,94], leaching of surface-applied solutes is also increased with high intensities of rainfall. Although the study outlines research scope for several topics on macropore water flow and solute transport, it presents the fact that the mechanism is uncertain, and the flow configuration or geometry is very variable and will always have to be estimated.

Ref. [95] conducted a review of modern methods of performing soil column experiments for both unsaturated and saturated columns. That study discussed disadvantages and advantages of different experimental methods, including best practices that can potentially solve column design problems in undisturbed monolith-type and repacked soil columns. Four soil column types were presented; viz., unsaturated, saturated, monolith, and packed soil columns. While the unsaturated and saturated types of soil columns are used to address issues regarding saturation levels, the discussions of monolith and packed soil columns are concerned with the method of construction of the soil columns.

According to [96], packing of the soil columns for laboratory experiments promotes homogeneity and avoids preferential flow. Packing methods for soil columns laboratory experiments include damp packing [97,98,99,100] which involves mechanical packing of small amounts of damp or dry soils into the soil column, slurry packing [101,102,103] which includes settling the saturated soil at the bottom of the column, and other methods such as wetting and drying cycles to assist compaction [104].

### 3.2. Single Porosity Models

The conventional Richards’ equation for unsaturated or variably saturated flow and the convection–dispersion equation for solute transport are typically the foundation for macroscopic models of water and solute movement in soils [52,62,63].

The Richards equation governing water dynamics in the unsaturated zone and the advection–dispersion equations (ADE) are as follows [105]:
(3)$$\frac{\partial \theta }{\partial t}=\frac{\partial }{\partial z}\left\lfloor K\left(\frac{\partial h}{\partial z}+1\right)\right\rfloor -S$$
(4)$$\frac{\partial \left(\theta C\right)}{\partial t}=\frac{\partial }{\partial z}\left\lfloor \theta D\frac{\partial C}{\partial z}\right\rfloor -\frac{\partial q\theta C}{\partial Z}$$
where t is time [T], z is depth measured positive downwards from the land surface [L], K is the soil unsaturated hydraulic conductivity [LT^−1^] as a function of h or θ, θ is the volumetric water content [L^3^L^−3^], S is a sink term (L^3^L^−3^T^−1^), h is the soil water pressure head [L] or the matric potential, which is of negative value in unsaturated soils, D is the effective solute dispersion coefficient [L^2^T^−1^], C is the solute concentration in liquid phase (ML^−3^), and q is the volumetric fluid flux density [LT^−1^] provided by Darcy’s law as follows:
(5)$$q=-K\frac{\partial h}{\partial z}+K$$

If the flow is steady, and the soil is homogeneous (q and θ are constant in time and space), Equation (4) is reduced to the 1D convection–dispersion equation:
(6)$$R\frac{\partial c}{\partial t}=D\frac{\partial^{2}c}{\partial z^{2}}-v\frac{\partial c}{\partial z}$$
where v = q/θ is the average pore-water velocity [LT^−1^]

The standard models, while still helpful for many applications, are unable to describe preferential flow in structured soil because, among other simplifying assumptions, they assume homogeneity and local-equilibrium conditions within a representative elementary volume (REV) (e.g., laminar flow, rigidity of the solid phase, no air phase effects). Decoupling the pressure head from the water content in the retention function, based on the single-domain model method, allowed [106] to simulate local non-equilibrium and produce an equation containing the independent variables θ and h. The real water content moving towards the equilibrium water content, θ_e_, is described by a kinetic equation as follows:
(7)$$\frac{\partial \theta }{\partial t}=f\left(\theta ,\theta_{e}\right):f\left(\theta ,\theta_{e}\right)=\left(\theta_{e}-\theta \right)/\tau$$
where τ is the equilibration time constant and a linear driving function is applied. The Richards equation ‘s numerical solutions can simply adopt this strategy. With very little adjustments to the water content, this enables the simulation of the swift passage of non-equilibrium moisture fronts through the soil as indicated by significant changes in pressure head. In order to maintain numerical stability and mass balance, the Richards’ equation is discretised spatially using the numerical manifold method (NMM) and temporally using the backward Euler scheme, which introduces under-relaxation and mass lumping techniques [107]. Ref. [108] discretised the Richard’s equation, both geographically using a finite element method and temporally using an implicit Euler scheme. To maintain the stability of the numerical simulation, mass-conservative and mass-lumping techniques were employed. With very little adjustment to the water content, this enabled simulation of the swift passage of non-equilibrium moisture fronts through the soil, as indicated by significant changes in pressure head.

### 3.3. Dual Porosity Models

According to dual-porosity models, which limit water flow to fractures, water in the matrix or intra-aggregate pores of the rock matrix is assumed to be immobile. Intra-aggregate pores are stationary pockets that do not allow convective flow but can interchange, retain, and store water. This conception results in two-region dual-porosity type flow and transport models [52,109]. This divides the liquid phase into zones that are static (stagnant, intra-aggregate), θ_m_, or mobile (flowing, inter-aggregate), θ_f_, expressed as follows:
(8)$$\theta =\theta_{m}+\theta_{f}$$
where subscripts f and m represent fractures, intra-aggregate pores, and macropores, and the soil matrix, intra-aggregate pores, and rock matrix, respectively. The Richard’s Equation (3), which describes water flow in the cracks, and the mass balance equation, which describes moisture dynamics in the matrix, can be combined to generate the dual-porosity formulation for water flow, as follows:
(9)$$\frac{\partial \theta_{f}}{\partial t}=\frac{\partial }{\partial z}\left[K\left(h\right)\left(\frac{\partial h}{\partial z}+1\right)\right]-S_{f}-\Gamma_{w}\;\frac{\partial \theta_{m}}{\partial t}=-S_{m}+\Gamma_{w}$$
where $\Gamma_{w}$ represents the rate at which water moves from the inter- to the intra-aggregate pores; S_f_ and S_m_ represent sink terms in both zones.

### 3.4. Dual Permeability Models

According to dual-permeability models, the matrix and the fracture pore domain are two distinct interacting subsystems that can represent the entire porous media system. The fracture domain in certain soils, such as cracked clays, may be empty, which causes a major difference in the physical behaviour of the soil compared with capillary flow. When a system has two porous continua, or dual porosity, it means that flow can occur in both porous domains, which are distinguished by different hydraulic conductivities. This is referred to as dual permeability. These kinds of models vary primarily in how they describe mass transfer between the matrix and fracture pore domains and flow within the macropore or fracture pore system. The flow equations for the fracture (subscript f) and matrix (subscript m) pore systems are as follows [52,59,110,111]:
(10)$$\frac{\partial \theta_{f}}{\partial t}=\frac{\partial }{\partial z}\left(K_{f}\frac{\partial h_{f}}{\partial z}+K_{f}\right)-S_{f}-\frac{\Gamma_{w}}{w}$$
and
(11)$$\frac{\partial \theta_{m}}{\partial t}=\frac{\partial }{\partial z}\left(K_{m}\frac{\partial h_{m}}{\partial z}+K_{m}\right)-S_{m}+\frac{\Gamma_{w}}{1-w}$$
where w is the ratio of the volumes of the fracture (inter-aggregate) and the total pore systems, θfsθs.

### 3.5. Multiple Porosity/Permeability Models

Though they have extra overlapping pore regions, multiple-porosity and permeability models essentially resemble dual-porosity models. More flexibility is thus possible, but at the cost of necessitating additional criteria that might also be inadequately specified physically.

Based on the assumption of a unit hydraulic gradient and a piecewise linear approximation to the hydraulic conductivity function, ref. [112] suggested a multi-domain model of solute mobility. Before redistribution, water fractions and solutes from each pore class were mixed in a single pool to compute the solute exchange between pore classes. Ref. [58] assumed three overlapping pore regions: primary fractures, secondary fractures, and soil matrix (i.e., macropores, mesopores, and micropores), while ref. [113] developed the MURF and MURT models for multi-region flow and transport, respectively. The multi-region model TRANSMIT [113] was built based on the single-domain model LEACHM. This model considers an overlapping pore zone and again uses the Richards and convection–dispersion equation to describe the flow and transport in each region. Like [58], they used convective and first-order diffusive transfer for solutes and first-order mass-transfer terms for water to enable water and solute exchange throughout all pore areas.

## 4. Experimental Measurement of Soil Hydraulic Properties

The understanding of soil hydraulic properties is crucial for many environmental science applications, such as (i) diagnosing the hydrodynamic functioning of soils in relation to the applied natural and/or human constraints, and (ii) simulating physical processes to establish a prediction on the order of magnitude of the hydraulic fluxes capable of, for example, providing nutrients and water to plant rooting systems or advecting chemicals that lead to diffuse pollution of the groundwater table [114]. The established paradigm was put to the test in the 1950s to early 1970s by fresh experimental observations of fast non-equilibrium water flow in macropores and the ensuing impact on solute displacement patterns. To determine soil parameters like temperature, water content, and soil pore pressure as well as functions like soil hydraulic conductivity function, soil water characteristics curve, and soil water diffusivity, experimental measurements of soil hydraulic properties are conducted using soil columns, either outdoors or in a laboratory equipped with instrumentation. This is typically accomplished by enclosing the soil column, for structural reasons as well as to stop fluid loss, in an impermeable and stiff shell material.

According to [95], since 1950, a great deal of work in the domains of hydrology, agriculture, and soil sciences has been published, most of which relies on the findings of soil column experiments. Still, there has never been an attempt to standardize or gather the best practices for building soil columns, and a survey of the literature reveals a dizzying variety of technical methods. A few of the tiniest columns documented in the literature have a diameter of 1 cm and a length of 1.4 cm [115]. The largest, however, weigh more than 50 tonnes and measure up to 2 m × 2 m × 5 m [116]. Historically and generally speaking, soil columns that function in the unsaturated regime have been called lysimeters [117], as shown in Figure 2, below. Large outdoor soil columns are typically referred to by this phrase, even though there is no specification that states a minimum size. These columns are commonly used to replicate conditions found in soil between the earth’s surface and the top of the groundwater table, also known as the vadose zone or unsaturated zone. These columns are characterized by having both air and water in their pore spaces. On the other hand, the pore spaces of soil columns operating in the saturated regime are devoid of any gaseous phase or air. In this case, a liquid such as water or an oil that is not in the aqueous phase fills the pores completely. Usually, these soil columns are employed to mimic the environmental parameters of an aquifer. There are significant design variations in the soil columns used to simulate saturated and unsaturated conditions.

Soil columns can be categorised based on two factors: the building method or the saturation level, as previously described. The literature has documented two main types of construction: monolith columns that employ undisturbed soil and packed columns that use disturbed soil. Excavated or “disturbed” soils are used to construct packed soil columns. The dirt is then backfilled into a rigid container and compacted. In contrast, monoliths are taken out of the natural soil whole and undamaged. Depending on the goals of the experiment, packed columns may or may not be preferred over monoliths because of their greater homogeneity. It has been demonstrated that the experimental outcomes directly depend on whether packed or monoliths columns are used.

### 4.1. Field Measurement of Soil Hydraulic Properties

Numerous methods have been put forth for the in situ determination of saturated hydraulic conductivity, Ks, via infiltration measurements. The undisturbed soil core method (SCM), rainfall simulator (RS), single- and double-ring infiltrometers (SRI and DRI), tension permeameter (TP), constant-head well permeameter (CHP), and falling-head borehole permeameter are some of the frequently used instruments. However, results from in situ measurements of saturated hydraulic conductivity, Ks, using commonly used tools and methodologies have often been inconsistent. The double-ring infiltrometer (DRI), the Guelph version of the constant-head well permeameter (GUELPH-CHP), and the CSIRO version of the tension permeameter (CSIRO-TP) were the three traditional devices whose Ks estimates were compared by [118], as shown in Figure 3. By using steady deep flow rates obtained from controlled rainfall-runoff experiments as benchmark values of saturated hydraulic conductivity, Ks, at local and field-plot scales, researchers assessed these methods’ capacity to reliably yield repeatable values and to detect the plot-scale variation of Ks. Two rings with a diameter of 30 cm on the inside and 55 cm on the outside were employed for the DRI (Figure 3a) measurements. These were sunk at least 5 cm deep and filled with water to almost the same level in both rings in order to create a virtually one-dimensional flow underneath the inner ring, where the infiltrated water depth at successive time steps was tracked. A bubbling tower, a graded water reservoir, and a 20 cm radius disc were used to perform the positive head CSIRO-TP measurements (Figure 3b). The GUELPH-CHP equipment (Figure 3c) was inserted into a borehole that was 8 cm in diameter and 15 cm deep.

It was observed that the three devices’ estimates of Ks were not very accurate when compared to benchmark values. The DRI overestimates by a factor of two in a laboratory context and by a factor of five at the plot scale. The GUELPH-CHP yielded significant variances in both cases, whereas the CSIRO-TP yielded better controlled overestimates but surprisingly significant regional heterogeneity of Ks in the lab soil. The researchers state that the causes of the observed discrepancies still need be looked into before using these techniques to confidently assess both field variability and local observations.

In Zaragoza, Spain, ref. [119] created a mobile, modified hood infiltrometer (MHI) design that allows the hydraulic properties of soil to be inferred from the transient cumulative infiltration curve. The MHI consisted of a water supply reservoir that is connected to a hat-shaped base that is placed on the soil’s surface, as shown in Figure 4. Corresponding values from the transient infiltration curve analysis were compared to the hydraulic conductivity (Ks) determined in a loam soil using the multiple-head approach and MHI. Following the MHI’s testing on three different soils under saturated conditions, the sorptivity (S) and Ks ascertained by the transient infiltration curve analysis were contrasted with the equivalent values acquired using a disc infiltrometer (DI). The authors claim that the data demonstrate that this approach makes it possible to obtain accurate estimations of both sorptivity and hydraulic conductivity. They added that studies showed that the prototype enabled precise computations of the soil hydraulic parameters on covered soil surfaces.

### 4.2. Soil Column Laboratory Measurement of Soil Hydraulic Properties

Ref. [120] created a novel permeameter that employs the steady-state method to calculate the hydraulic conductivity of unsaturated soils and directly detects suction (negative pore-water) values. In order to directly measure suction during the tests, two tensiometers are mounted in the apparatus, as shown in Figure 5. The instrument can be used to determine the hydraulic conductivity function of sandy soil across a low suction range of 0–10 kpa. Similar results were obtained from tests conducted on two identical sandy soil specimens from Edosaki and Chiba, Japan, which validated the measurement of unsaturated hydraulic conductivity using the innovative permeameter. Following drying and soaking procedures, the hydraulic conductivity functions of the two soils were ascertained. The measured unsaturated hydraulic conductivity functions were compared with the predictions using the Fredlund equations [8], the Van Genuchten estimation [121], and the Brooks and Corey estimation [122]. The results suggest that the different prediction methods could adequately explain the observed behaviour; nevertheless, the method of [8] yielded a more accurate forecast.

Ref. [123] recently developed a hydrogeophysical soil column system to monitor crucial hydraulic and electrical properties of regolith in critical zones. A cylindrical cell to hold soil samples and a special hydrogeophysical sensor to monitor electrical potential and pore water pressure in soils make up soil column system. The technology can simultaneously measure the fundamental hydraulic and electrical characteristics of unconsolidated materials in both saturated and unsaturated conditions. The generated soil column was tested using a sand sample mechanically packed in a cylindrical cell, collected from a riverbank close to Mores Creek in the United States of America, as shown in Figure 6. The results show that the saturated flow test from the system can be used to directly measure the saturated hydraulic conductivity Ksat, saturated complex electrical conductivity σ*sat, and saturated streaming potential coupling coefficient Csat. At the same time, the saturated flow can generate transient responses of pore water pressure, outflow, and self-potential SP, which can be processed to estimate the soil’s key hydraulic and electrical properties. Regrettably, the current design only allows consideration of reconstituted samples. Because the material structure can significantly affect the hydraulic and electrical characteristics of geological materials, caution should be used when interpreting geophysical field data using values acquired from reconstituted samples. Because some critical zone materials have many fine grains and the associated investigations take longer than the exhibited sand samples, it is also crucial to ascertain whether the setup remains successful for materials that are rich in clay or silt. Nonetheless, the arrangement that was constructed marks a significant advancement in the study of the petrophysical properties of minerals in the critical zone.

Over the course of 62 days in the lab, ref. [124] examined the flow of water in an unsaturated soil column that underwent many infiltration occurrences. The setup is shown in Figure 7. A 0.236-m-diameter, 4 m tall column supported sensors for measuring the water content and suction while the dirt was kept in place. Five water potential probes and fourteen moisture probes were installed in order to measure suction and water content. Soil with a somewhat higher natural groundwater table (GWT) and a disturbed hydraulic condition from decades of farming was taken from the Heifangtai loess platform in Gansu Province, China, and used in the experiment.

Clay (<0.002 mm) made up around 5.6% of the soil, whilst silt (0.002–0.05 mm) made up 84%. The results indicate the establishment of two wetting fronts, i.e., wetting front I and wetting front II, caused by the first and subsequent infiltration episodes, respectively. A stable zone with a relatively consistent water content developed in between the two fronts. A conceptual model of the suction profile that divides the unsaturated zone into four zones—the active, stable, transition, and capillary fringe zones—was developed in order to comprehend the in situ water flow. The way water moves through different zones is logically explained by this classification. Additionally, 1D numerical analysis was carried out to investigate the flow and expand the seepage theory to unsaturated soils. Figure 8 shows the soaking SWCC of compacted loess. There are three components to the figure: (i) measured data from the axis translation technique; (ii) the fitted curve using van Genuchten’s equation [122] and related fitting parameters; and (iii) data from the column test using moisture probes and water potential.

Ref. [125] developed an automated soil water retention test system that uses a computer to remotely control the entire process and measure and record volume changes during testing to determine the soil water retention curve. The capacity of the new system to automatically assess the wetting and drying qualities with high precision and measure volume change throughout testing using a single sample are only two of its many notable advantages over existing methods, according to its developers. Four different soil types—K-8, Takeda, Sasaguri, and Fukuchi, all gathered from Japan and with textures varying from sandy to silty—had their water retention curves calculated while taking volume change into consideration. Figure 9 shows how volume variation affects the soil water retention curves. Since the volumetric water content cannot be determined directly, the volume change of soil specimens has an additive influence on the volumetric water content or the degree of saturation. This indicates that when the slope of the soil water retention drops, the influence of volume change on soil water retention curves increases in the residual condition. The system was reportedly limited by the use of acrylic acid resin in the main unit, which limited the suitable air pressure to about 300 kPa. In light of this, the test system is suitable for studies involving sandy and silty soils. It is not appropriate for clay soils.

Ref. [126] carried out irrigation experiments on a sizable undisturbed volcanic soil column and suggested a method to analyse the viability of various measurement methodologies to determine the flow parameters from transient flow data. Data on soil water content, bottom flow, and/or matric pressure head were introduced in the inversion problem. The parameters were inversely estimated using the water flow module of the WAVE model in combination with the GMCS and NMS method (Global Multilevel Coordinate Search combined with a Nelder Mead Simplex, GMCS-NMS). A sizable monolith of undisturbed volcanic soil with a sandy–clay–loam texture was excavated from a field in Tenerife, Spain, using steel cylinders (θ45 × 85 × 0.4 cm). After that, it was brought to the lab and equipped with 21 TDR probes and seven digital tensiometers, as seen in Figure 10a, which were positioned at seven observation depths (designated as A–G). With a 550 × 550 × 32 mm^3^ plexi-glass box and 310 hypodermic needles (θ0.3 × 6 mm spaced 20 mm apart) to deliver water evenly at the top of the column, a small rainfall simulator was constructed. The findings describe how the water retention information at each soil depth from the initial irrigation experiment was used to select the parameters for the inverse modelling for the second trial.

The first experiment showed heterogeneities in the soil profile, as seen in Figure 10b, where four strata (H1–H4) with different water retention behaviours were identified.

In order to determine the level of model complexity needed to account for the experimental evidence of preferred flow in the microporous soil, ref. [127] developed a novel soil column experiment using state-of-the-art measuring techniques. These methods also made it possible to distinguish between macropore and matrix water flow and quantify interdomain (macropore–matrix) water transfer. The pseudo-three-dimensional axisymmetric Richards’ equation (ARE) was solved inversely to determine the parameters for the single- and dual-domain preferred flow models. The new experimental setup, shown in Figure 11a, consisted of an acrylic column that was 24.4 cm in diameter and 80 cm long. It included a perforated PVC plate with two holes of 0.3 cm per cm^2^ at the bottom, covered with a nylon membrane. Separate exits for macropore and matrix effluent were also present, along with a central cylindrical flow divider to lessen lower boundary effects on the water exchange flux between the two. Coarse sand was placed in the annulus while dry-sieved, graded sandy loam from a field near College Station, Texas, was placed in the column.

After establishing the matrix and macropore domains, the system was instrumented with tensiometers and temporal domain reflectometry (TDR) probes to record the water content and pressure head at various depths. The hydraulic functions for the matrix and macropore domains, which were generated by fitting the upward infiltration and infiltration drainage data using the axisymmetric Richards’ equation (ARE) technique, are shown in Figure 11b. The figure illustrates the notable differences between the hydraulic properties of a matrix and a macropore. The findings demonstrate that local hydraulic non-equilibrium of pressure heads, water contents, and flow velocities in matrix and macropore domains are characteristics of preferential flow.

In a study on sandy soil in a lab setting, ref. [128] estimated the development of soil pore water electrical conductivity (σp) over time using a time-varying dynamic linear model (DLM) and the Kalman filter (Kf). A time series of the relative dielectric permittivity εb and (σb) of the soil were measured using time domain reflectometry (TDR) at different depths in a soil column in order to transform the deterministic Hilhorst model into a stochastic model and evaluate the linear relationship between the relative dielectric permittivity εb and the bulk electrical conductivity (σb) in order to capture the deterministic changes to (1/σp). Sand was added to the columns with a density of 1.4 g/cm^3^ and a water content of about 4 m^3^/m^3^. TDR and soil temperature sensors were positioned at four distinct depths to collect enough data for modelling. Every five minutes, the soil relative dielectric permittivity (εb), bulk electrical conductivity (σb), and temperature were measured (Figure 12). Using the Hilhorst model, the authors report detecting strong positive autocorrelations between the residuals. Furthermore, it has been reported that by applying and modifying them to DLM, the observed and modelled data εb obtained a significantly better match, and the anticipated development of (σp) converged to its correct value.

As illustrated in Figure 13, ref. [129] evaluated the hydraulic characteristics of unsaturated soil in the lab using a real-world soil column experiment. A specially designed cylindrical vertical soil column rig was built; a WET150 sensor by DELTA-T Devices Limited (UK) was used to measure the volumetric soil water content θ (%) and an EQ3 Equitensiometer sensor by the same company was used to measure the soil matric potential Ψ (kPa). A GP2 data logger by DELTA-T Devices Limited (UK) was also linked to the sensors, and a computer that was directly connected to the data logger was used to output the results. The link between the volumetric soil water content and the soil matric potential was used to create the soil water characteristics curve. Two separate monoliths of undisturbed soil samples from Ivrogbo and Oleh in the Nigerian inland valley of the Niger Delta, as well as a uniformly packed sample of soil from Aberdeen, UK, for comparison, were subjected to gravity-driven flow experiments. According to reports, the sensors immediately recorded the soil’s electrical conductivity in mS·m^−1^, temperature in degrees Celsius, matric potential in KPa, and water content in percentage (%), as indicated in Table 2. However, the wall effect, experimental non-repeatability, and the need for sophisticated equipment presented technical hurdles.

A summary of the literature reports on the flow through variably saturated soils is presented in Table 3, while Table 4 presents a summary of the various experimental methods that have been reviewed with a focus on the measurable parameter, accuracy, cost, and applicable soil types.

## 5. The Future of Smart Sensors in Soil Column Water Flow Experiments

This analysis makes clear that several investigations into various aspects of flow through variable-saturated soils have been conducted in an effort to determine the soil’s hydraulic characteristics. Previous findings were dominated by modifications of Darcy’s equation. The literature provides model equations and model descriptions of the processes involved in water flow through saturated and unsaturated soils, as shown in Table 3. These comprise studies of models, such as single-porosity models, dual-porosity models, and dual-permeability models, that describe different types of flow in the matrix and macropore or fracture system. Methodologies for preferred flow models are also covered. The reviews discuss several concepts that characterise the hydrodynamic flow properties in subsurface soil, such as soil homogeneity and heterogeneity and whether the flow is uniform, preferential, or non-equilibrium. Through the numerical modelling of the Richards equation, great progress has been made in the last ten years in the construction of complex numerical models for flow through variably saturated soils. However, there are still a number of unanswered questions regarding the numerical models’ robustness, accuracy, and dependability across their wide range of applications. Additionally, in order to benefit from new computer designs and ongoing advancements in computing technology, there is an increasing demand for new numerical approaches, techniques, and algorithms.

Model description, model equations, and model creation have been the subject of a good sixty years’ worth of writing, beginning in 1964. The literature that was consulted for these reviews spans the past 25 years. Although there has been progress, ref. [62] reports that in order to address conceptual problems relating the preferential flow domain, measurement methods and methodologies for independent parameter determination are still required. Additionally, flow configuration or shape is highly variable and will always be estimated, and the macropore water flow mechanism is unknown [74]. The experimental determination of hydraulic parameters by field or laboratory measurements is also covered in the literature review. Studies on experimental measurement, both in the lab and in the field, have increased over the past ten years. Along with the soil matric potential, soil water diffusivity, soil hydraulic conductivity, soil pore water electrical conductivity, and soil water content, hydraulic characteristics also include the soil water characteristic curve, sometimes referred to as the soil water retention curve. Since 1950, a substantial quantity of research has been published in the domains of soil sciences, hydrology, and agriculture, most of which is based on the findings of soil column experiments [95]. A survey of the literature reveals an astonishing variety of technical approaches, but despite this, no attempt has ever been made to compile or standardise the best practices for generating soil columns. Determining the soil water characteristics curve has also been the topic of extensive investigation. Inconsistent results have frequently been obtained from in situ measurements of Ks obtained via widely used instruments and techniques.

However, in the upcoming years, research on a variety of soil-related topics will increase due to technological developments and more precise measurements of many characteristics [130,131]. In order to accurately quantify hydrological processes, identify preferential flow paths, comprehend the impact of water on soil properties, and validate hydrodynamic models with empirical data, the use of sensors in soil column experiments is crucial because these allow recise real-time monitoring of changes in soil water content, water flow, and pore water pressure.

Significant technical advancements are being adopted to improve decision-making in the analysis of soil properties [132]. Future soil column water flow experiments are likely to follow the trend of smart sensor technologies, utilizing sophisticated sensing devices with the ability to gather, process, and send data. These days, the best instruments for collecting soil properties and evaluating data in accordance with design-of-experiment principles are drones and sensors mounted on the ground. Smart precision technologies are used as wireless systems networks (WSN), Internet of Things (IoT)-based systems, remote sensors (RS), unmanned aerial vehicles (UAVs) or drones, and on-the-go sensors [133]. Over the past decade, an increasing number of sensor networks and technologies, including wireless sensor networks (WSNs) and Internet of Things (IoT)-based information systems, have been developed to give more precise and real-time data on soil parameters including soil water content. For example, IoT-based devices have been created for in situ measurements of soil moisture and other parameters. By integrating sensor networks with IoT platforms, automated control systems, data analysis through artificial intelligence (AI) and machine learning (ML), and remote monitoring are made possible. Application improves operational effectiveness and data accuracy. Proximity sensing has replaced conventional in situ sample collecting and lab oven-drying techniques in remote sensing (RS) in order to meet the demand for image processing, particularly in the past ten years. Installing many sensors or cameras in UAVs or drones enables proximity sensing. However, while precision sensor technologies offer significant benefits, they face challenges including high initial investment costs, complexities in data management, the need for technical expertise, data security and privacy concerns, and issues with connectivity in remote areas. To obtain more precise measurements, it is obvious that the scientific and user communities need to conduct more research on operational networks.

## 6. Conclusions

The Richards equation, a highly nonlinear partial differential equation, describes flow in a porous material that is variably saturated. With a focus on multiple porosity and permeability models, dual porosity models, single porosity models, and dual permeability models, this study examines water flow mechanisms across variably saturated soils. In preferential flow pathways and slow or stagnant pore regions, the models primarily attempt to characterise flow and transport independently. However, the majority of attempts to resolve saturated–unsaturated flow issues rely on numerical methods. The bigger issue is to identify and measure the soil-structural properties experimentally, even though there are still conceptual issues with the description of flow in the preferential flow domain. Thus, in addition to functions like soil hydraulic conductivity function, the soil water characteristics curve, and soil water diffusivity, the study also examines experimental measurements of soil hydraulic factors like temperature, water content, and soil pore pressure. Soil columns are used for experimental measurements of soil hydraulic parameters, which can be carried out outside or in a lab with sophisticated equipment and sensors. Advanced precision smart sensor technologies are progressively replacing traditional laboratory measuring methods as the focus shifts to efficient performance outputs. These cutting-edge technologies provide economical solutions while also quickly and labour-efficiently determining soil moisture values. More significantly, by boosting resilience to climate variability, encouraging sustainable agricultural methods, and increasing water-use efficiency, these technologies support the objectives of climate-smart agriculture. IoT systems, UAVs, and mobile sensors are some of the technologies that help optimise irrigation and cut down on excessive water use. Although each of these technologies has advanced significantly, in order to obtain more precise readings, there is still a lot of work to be done and a lot of room for improvement in the research into the application of smart sensors in soil column investigations by users and scientific communities.

## Figures and Tables

**Figure 1 sensors-25-07027-f001:**
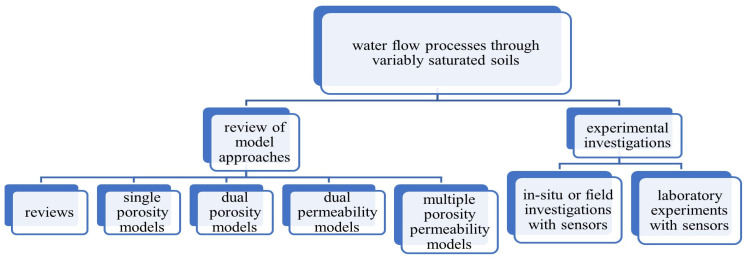
Water flow processes through saturated and unsaturated soils.

**Figure 2 sensors-25-07027-f002:**
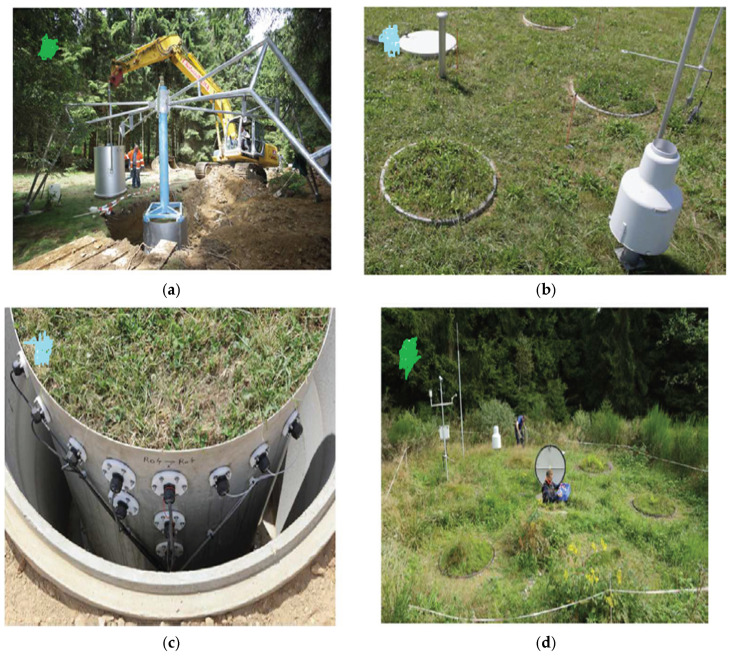
(**a**) Coring of an undisturbed lysimeter core, (**b**) installed lysimeter, (**c**) instrumentation, and (**d**) view of a lysimeter setup with underground access [117].

**Figure 3 sensors-25-07027-f003:**
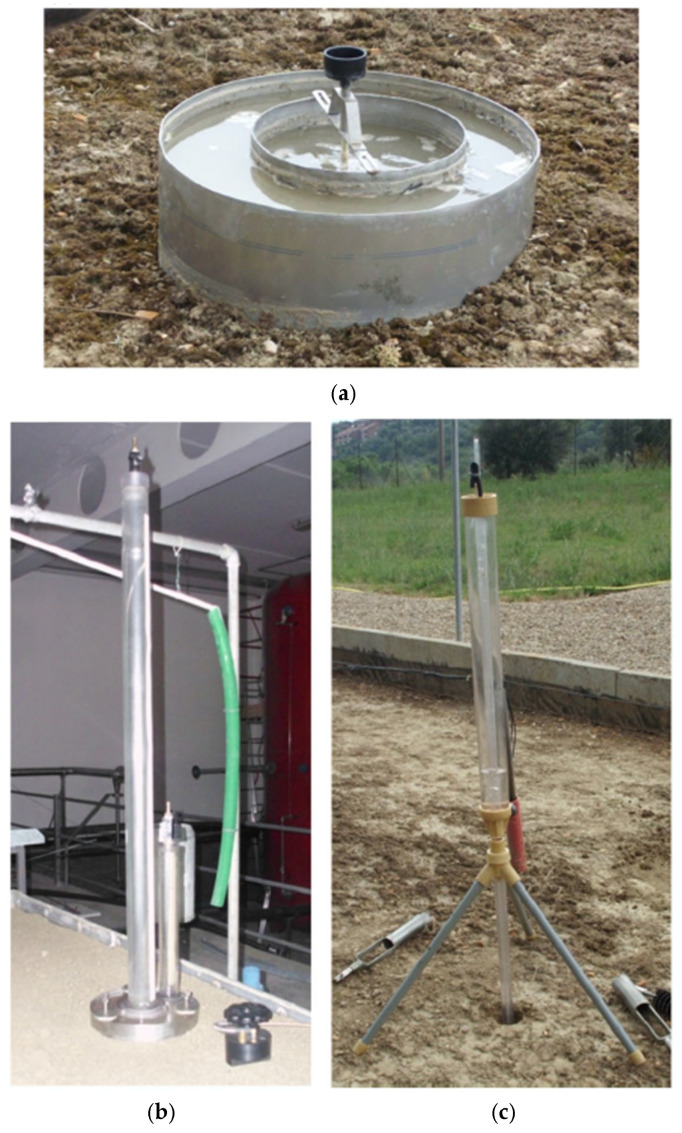
Estimation of soil hydraulic conductivity: (**a**) double-ring infiltrometer, (**b**) CSIRO positive-head tension permeameter, and (**c**) GUELPH constant head permeameter [118].

**Figure 4 sensors-25-07027-f004:**
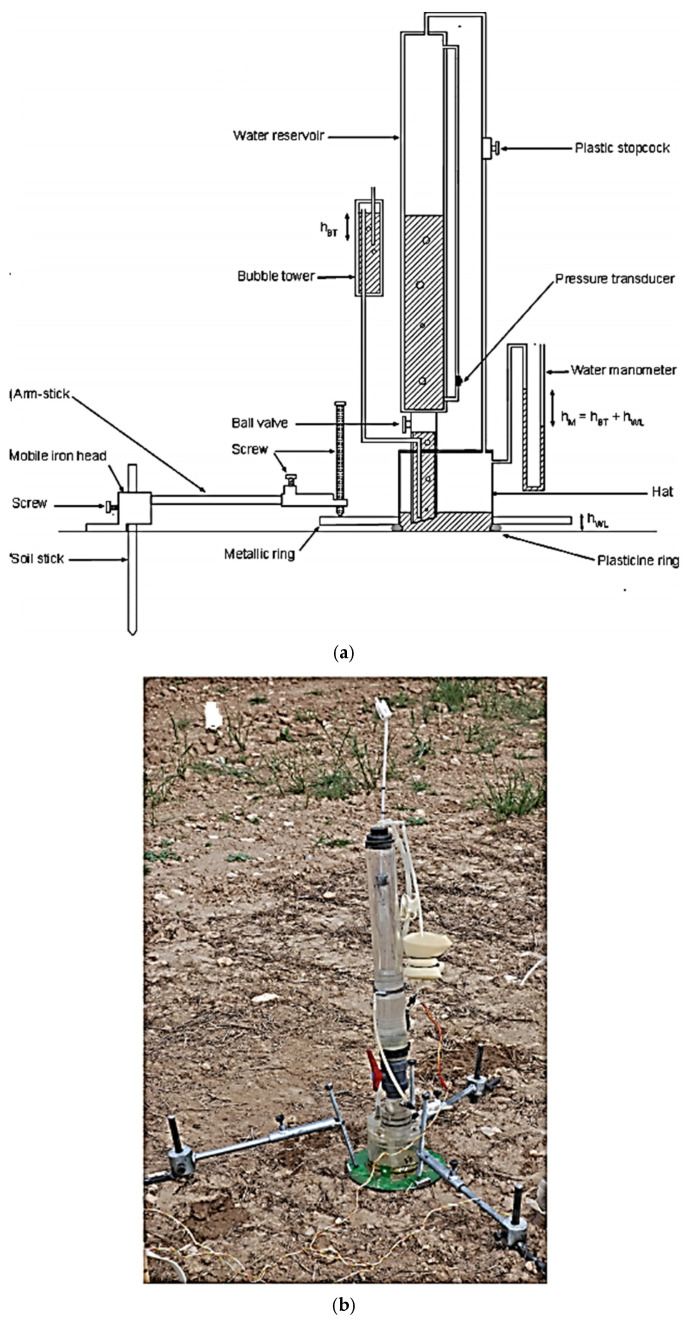
(**a**) Sketch and (**b**) picture of the modified hood infiltrometer [119].

**Figure 5 sensors-25-07027-f005:**
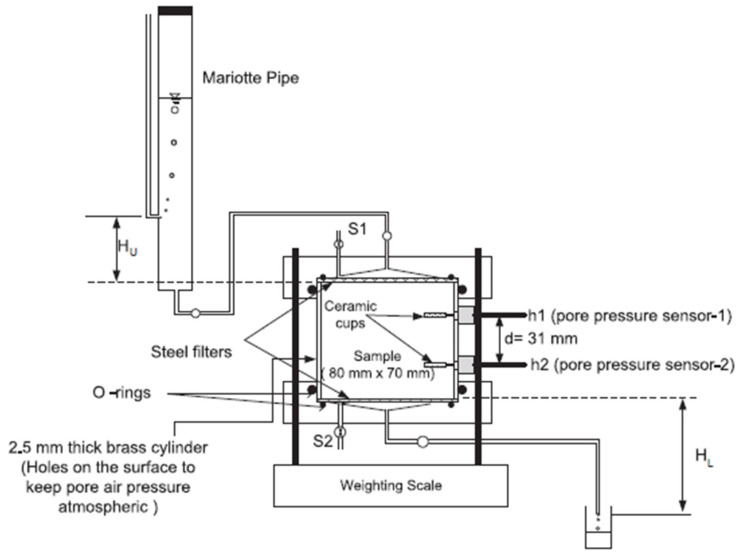
Novel permeameter for laboratory measurement of unsaturated hydraulic conductivity [120].

**Figure 6 sensors-25-07027-f006:**
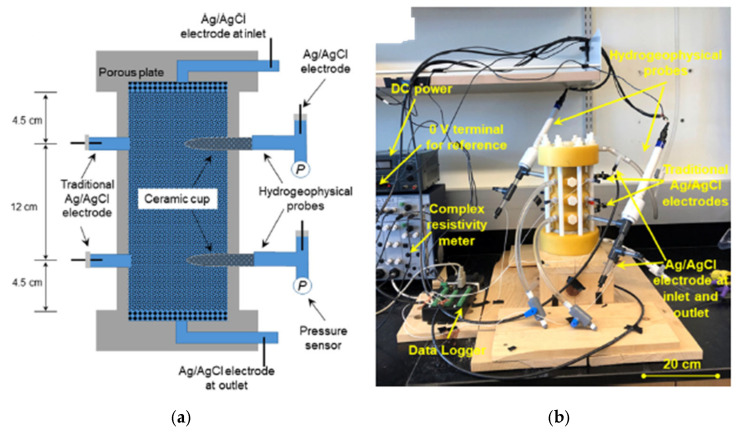
(**a**) The schematic and (**b**) soil column for novel hydrogeophysical probe [123].

**Figure 7 sensors-25-07027-f007:**
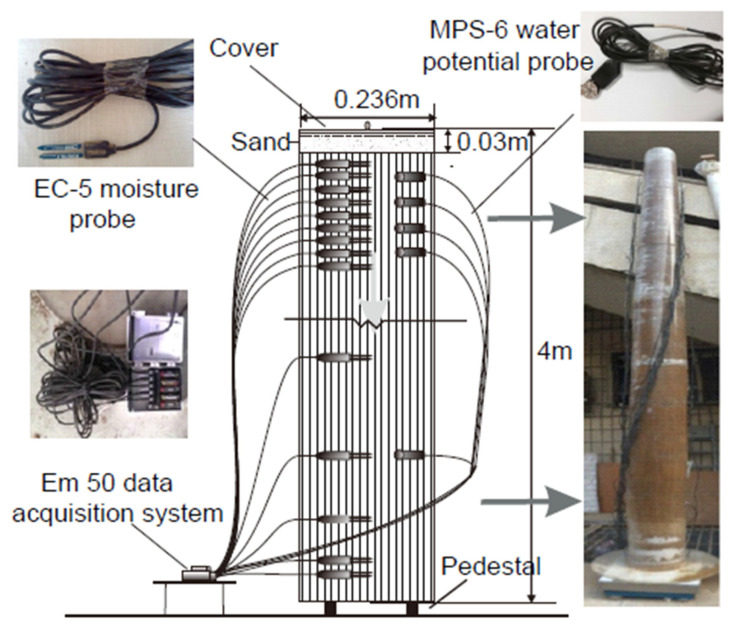
Soil column for water flow in unsaturated soil [124].

**Figure 8 sensors-25-07027-f008:**
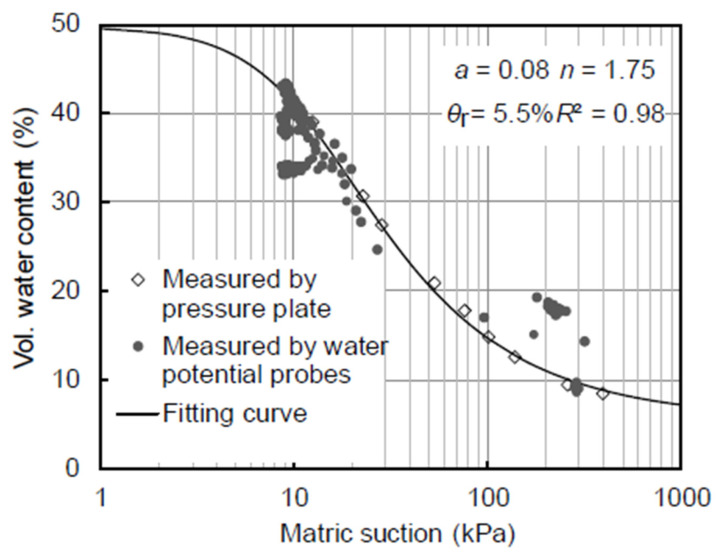
SWCC for the loess test [124].

**Figure 9 sensors-25-07027-f009:**
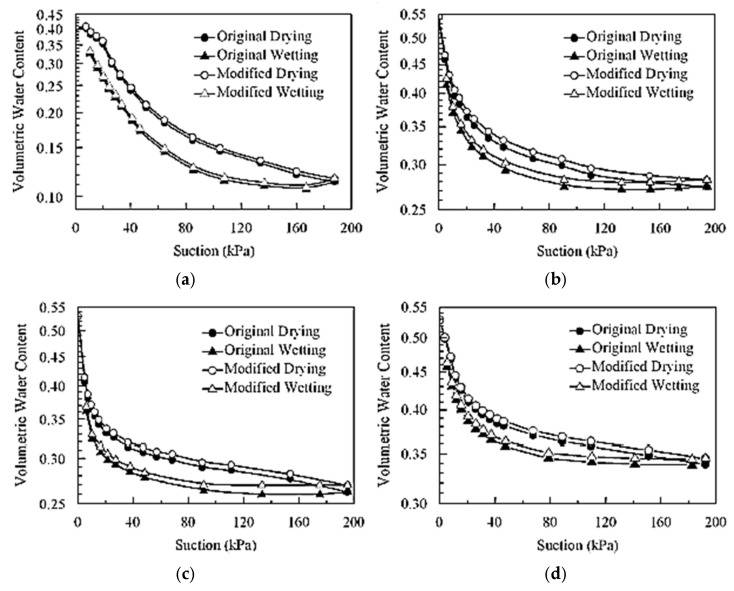
Effects of volume change on soil water retention curve: (**a**) K-8, (**b**) Takeda, (**c**) Sasaguri, (**d**) Fukuchi [125].

**Figure 10 sensors-25-07027-f010:**
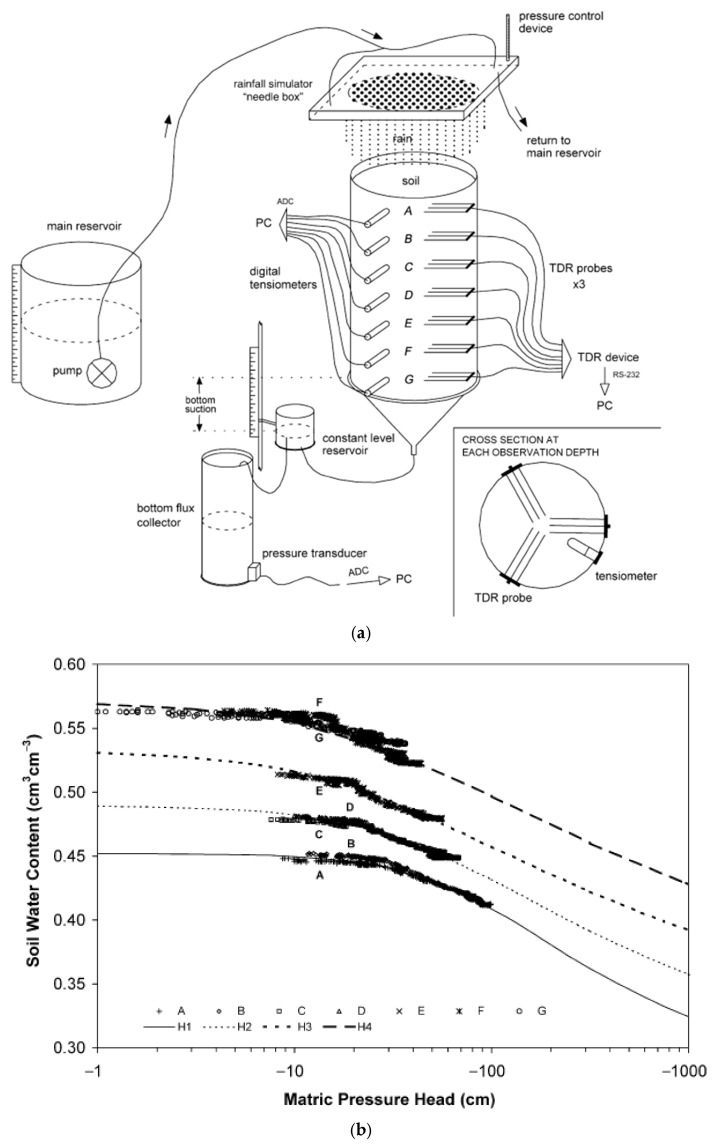
(**a**) Soil column laboratory set-up for volcanic soil monolith [126]. (**b**) Soil moisture retention curves observed in the monolith. Measured data (symbols) and fitted van Genuchten curves (lines) [126].

**Figure 11 sensors-25-07027-f011:**
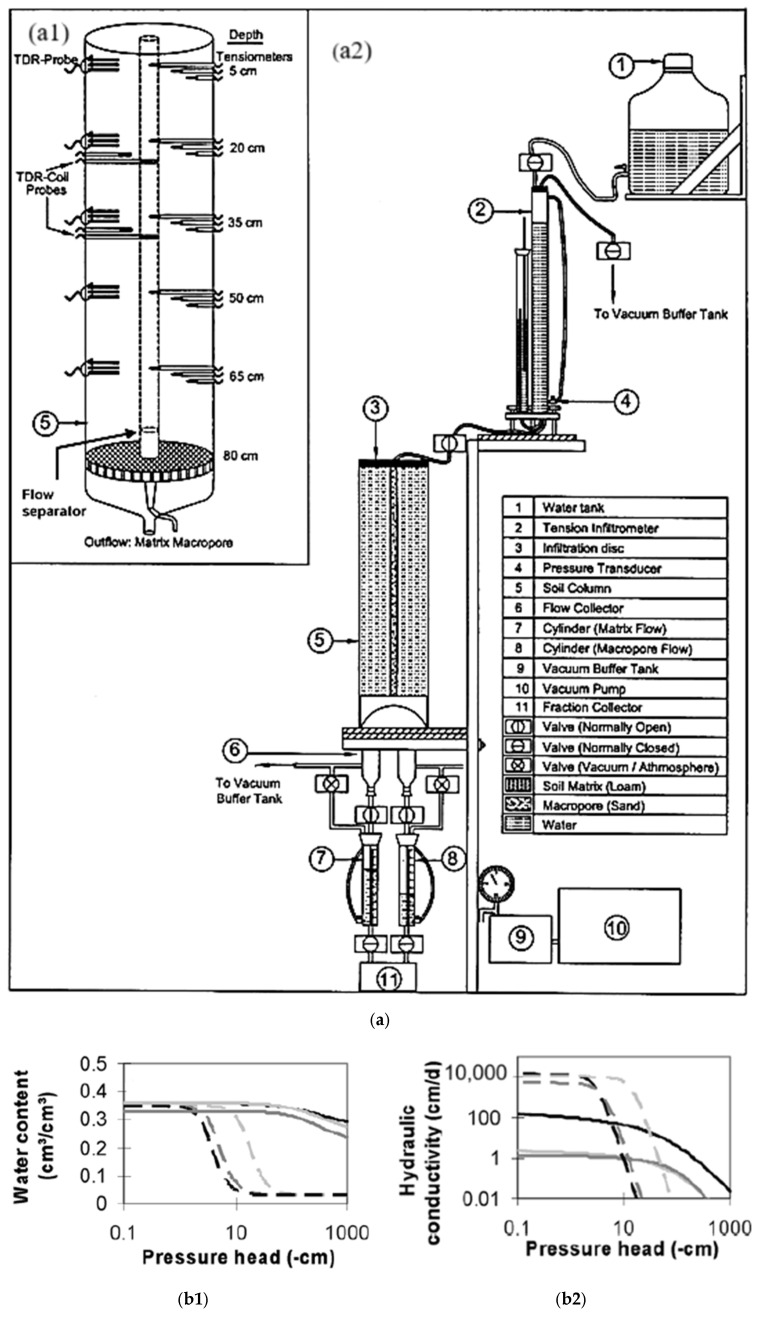
(**a**). Soil column experiment for studying macropore flow processes: (**a1**) column with instrumentation and (**a2**) complete setup [127]; (**b**) Hydraulic van Genuchten functions for matrix and macropore obtained by inverse parameter estimation of approach ARE for drainage, infiltration, and upward infiltration experiments: (**b1**) water retention and (**b2**) hydraulic conductivity [127].

**Figure 12 sensors-25-07027-f012:**
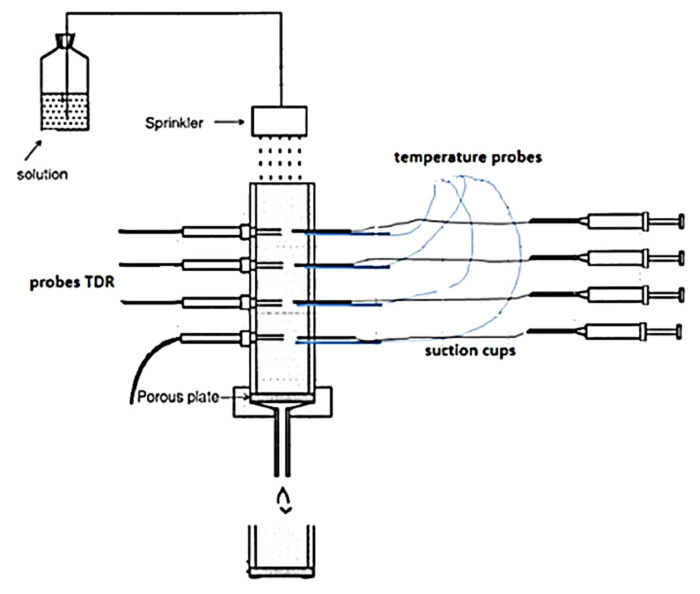
Soil column experiment for the determination of soil pore water electrical conductivity [128].

**Figure 13 sensors-25-07027-f013:**
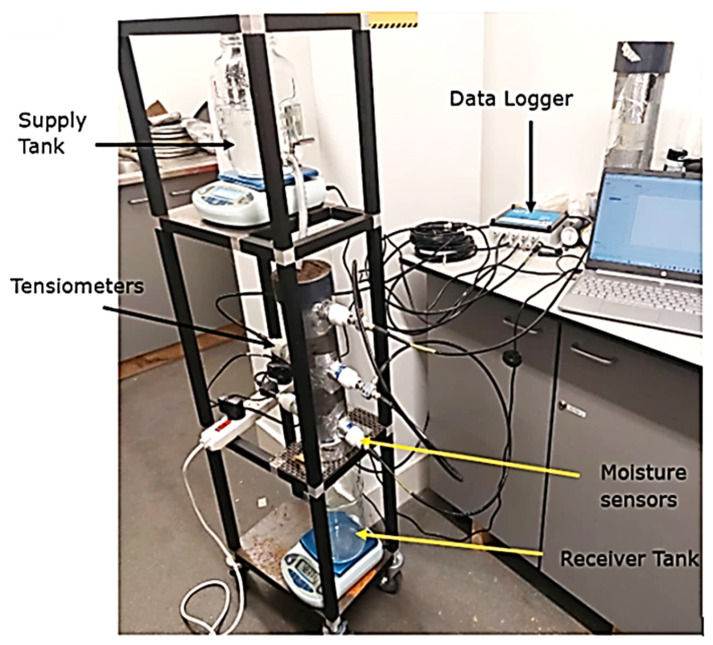
Real-world soil column experiment to assess hydraulic parameters of unsaturated soil in the laboratory [129].

**Table 1 sensors-25-07027-t001:** The finger radius (Rf) expressed theoretically using either dimensional analysis/experimentation or linear stability analysis [38].

No.	References	Expression for R_f_ [L]
1	[42,43,44]	Rf,1=4.16σρg121−PKf−12
2	[45,46]	Rf,2=4.16θf−θi−12σρg121−PKf−12
3	[43,44,47]	Rf,3=4.8R*hwe121−PKf−12
4	[44,48]	$R_{f,4}=\frac{2.4S^{2}}{K_{f}{\left(1-\frac{P}{K_{f}}\right)}^{\left(\theta_{f}-\theta_{i}\right)}}$
5	[49]	$R_{f,5}=\frac{4.8\theta_{f}{\left(\frac{dy}{d\theta }\right)}_{\theta_{f}}}{\eta +1.5}$
6	[44,50]	$R_{f,6}=\frac{4.8\theta_{f}}{K_{f}\left(1-\frac{P}{K_{f}}\right)\left(\theta_{f}-\theta_{i}\right)}\int_{h_{i}}^{h_{f}}K\;dh$
7	[44,47,48]	$R_{f,7}=0.6\left|h_{we}-h_{ae}\right|{\left(1-\frac{P}{K_{f}}\right)}^{-1}$
8	[51]	Rf,8=2.05S2Kfθf−θi1−PKf12

**Table 2 sensors-25-07027-t002:** Recorded soil hydraulic parameters from GP2 data logger connected to WET 150 and EQ3 Equitensiometer sensors.

Psi	θ	SoilTemp	ECp
kPa	%	deg C	mS·m^−1^
−436.5	2.3	19.8	#-INF
−436.6	2.3	19.8	#-INF
−439.9	2.3	19.8	#-INF
−435.3	2.5	19.8	#-INF
−423.1	2.2	19.8	0

**Table 3 sensors-25-07027-t003:** Summary of the Literature Reports on Flow through Variably Saturated Soils.

S/N	References	Year	Research Activity	Parameters	Sensors	Findings	Location
1	[38]	2000	Review of finger flow models	Finger radius, R_f_	-	Eight expressions categorised into three groups	-
2	[52]	2003	Review of modelling preferential and non-equilibrium flow	-	-	Explanation of preferential flow in the structured media and prompting of development of Hydrus-1D and Hydrius-2D	-
3	[62]	2006	Review of model approaches to preferential flow	-	-	Flow through soil is uniform or nonuniform	-
4	[74]	2007	Review of non-equilibrium flow.	-	-	Mechanism of macropore water flow and solute transport is uncertain and the geometry is guessed	-
5	[95]	2010	Review of methods of conducting soil column experiments	-	-	Monolith and packed soil columns affect method of soil column construction	-
6	[96]	2007	Review of experimental methods	-	-	Packing of the soil column for lab exp. promotes homogeneity and reduces preferential flow	-
7	[63]	2004	Model description	C, K, D, h, θ, t, τ	-	Single-porosity, double-porosity models	-
8	[109]	1976	Model description	K, t, θ, θ_f_, θ_m_, S_f_, S_m_, $\Gamma_{f}$, $\Gamma_{m}$	-	Dual-porosity formulation for water flow	-
9	[111]	2023	Model description	K, t, θ, θ_f_, θ_m_, S_f_, S_m_, $\Gamma_{f}$, $\Gamma_{m}$	-	Dual permeability	-
10	[58]	1995	Model development	θ, S, D, ψ	-	MURF and MURT models	-
11	[121]	1980	Model equation	θ, S, D, ψ	-	SWCC, diffusivity, sorptivity	-
12	[8]	1994	Model equation	θ, S, D, ψ	-	SWCC, diffusivity, sorptivity	-
13	[122]	1964	Model equation	θ, S, D, ψ	-	SWCC, diffusivity, sorptivity	-
14	[129]	1991	Hydraulic properties description	θ, S, D, ψ	-	SWCC, diffusivity, sorptivity	-
15	[118]	2017	Field Experiment	θs, θi, q, ψ, Ks	DRI, GUELPH-CHP, CSIRO-TP	Devices’ estimates were not accurate	Perugia in Italy
16	[119]	2015	Field experiment	Ks, S	MHI	Accurate calculation of soil hydraulic properties	Zaragoza in Spain
17	[120]	2013	Laboratory experiment	Ks	Permeameter, Tensiometers	Fredlund forecast was more accurate	Specimen from Japan
18	[123]	2022	Laboratory experiment	Ks, σ*s, Cs	Hydrogeophysical probe	Direct measurement of hydraulic properties	Specimen from Mores Creek in USA
19	[124]	2019	Laboratory experiment	θ, ψ	Moisture probes, water potential probes	Wetting SWCC displayed.	Specimen from Gangsu Province in China.
20	[125]	2012	Laboratory experiment	θ, ψ	Oedometer-type device, control software, water pressure transducer	As slope of soil water retention curve decreases, effect of volume change increases	Specimen from K-8, Sasaguri, Fukuchi, Takeda in Japan
21	[126]	2004	Laboratory experiment	θ, ψ, q	Tensiometers, TDR probes	Water retention data from the first experiment were useful for selecting parameters for inverse modelling in the second experiment	Specimen from Tenerife in Spain
22	[34]	2005	Laboratory experiment	θ, ψ	Tensiometers, TDR probes	Hydraulic characteristics of macropore and matrix differ significantly	Specimen from College station in Texas.
23	[127]	2018	Laboratory experiment	σ_p,_ $\sigma_{b}$, $\varepsilon_{b}$	TDR probes, temperature probes	High positive autocorrelations between the residuals using the Hilhorst model.	Berlin, Germany
24	[128]	2025	Laboratory Experiment	θ, ψ, ECp	WET 150, EQ3 Equitensiomenter	Direct measurement of soil water content, soil matric potential, soil electrical conductivity	Specimen from Niger Delta area of Nigeria and Aberdeen in the UK.

**Table 4 sensors-25-07027-t004:** Comparison of various experimental methods.

Experimental Method	Measurable Parameter	Accuracy	Cost	Applicable Soil Type
Single-ring infiltrometer (SRI)	Ks	Limited.	USD 100–150	Sandy and loamy soils
Double-ring infiltrometer (DRI)	θs, θi, q, ψ, Ks	Limited by several factors.	USD 280–350 (lab) USD 1000–3000 (field)	Sandy, silty, loamy soils
Soil core method (SCM)	$\rho_{b}$, θ, C, EC	Affected by discrepancies.	Variable, depends on project scope	Loam, silt, clay, not coarse soils
Rainfall simulator (RS)	θ, ψ, $\rho_{b}$	Generally accurate to 80%.	USD 120–150 educational models	Sand, loam, silt, clay
Tension permeameter (TP)	ψ, q	Depends on soil properties.	USD 3000–10,000	All soil types
Constant-head permeameter (CHP)	θs, Q, ψ, Ks	Affected by soil type and permeability	USD 250–4000	Sand and gravels
GUELPH-CHP	θs, Q, ψ, Ks, S	Variable, though reliable for field estimate.	USD 5500–8600 full kit	Sand and gravels
CSIRO-TP	Ks, S	Variable accuracy	USD 800–3000	From coarse-textured sand to fine-textured clay loams
Tensiometer	ψ	±1% for wider range	USD 50–150 (simple soil tensiometers)	Sandy soils and other light-textured soils
TDR probes	T, θ, $\varepsilon_{a}$	2–3%	USD 1900 (complete soil moisture unit)	All soil types
Temperature probes	T	±0.1–1.5%	USD 30–300 (for industrial applications)	All soil types
WET 150	θ	±3%	USD 800	Mineral soils, organic soils, peat
EQ3 equitensiometers	ψ	±10%	USD 900	All non-saline soil types

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
