# Peer review of "Advances in Flow of Water Through Variably Saturated Soils: A Review of Model Approaches and Experimental Investigations with Use of Sensors"

_sensors, 2025, doi:10.3390/s25227027_

Round 1

Reviewer 1 Report

Comments and Suggestions for Authors

The manuscript is a review article that systematically summarizes the numerical models and experimental studies related to water flow in variably saturated soils, with a particular focus on the recent advances in determining soil hydraulic properties using sensors. The paper has a complete structure, clear logic, and rich references covering nearly sixty years of research progress. Its summary of soil hydrodynamics, model classifications, and sensor applications provides valuable insights. However, the paper could be further improved in terms of academic depth, systematic analysis, and conciseness of future perspectives. Some sections show overlapping content, repeated descriptions, and imprecise conceptual explanations, which affect the clarity and originality of the review.
1、Sections 3 and 4 contain numerous theoretical equations and repetitive explanations. It is recommended to merge them into a single chapter entitled “Review of Numerical Models,” with subsections discussing single-porosity, dual-porosity, dual-permeability, and multiple-porosity models. The relationships between parameter identification and experimental validation should also be emphasized.

2、Section 5 summarizes a large number of experimental methods (both field and laboratory) and related instruments but lacks horizontal comparison and technical evaluation. It is suggested to add a table or matrix comparing the advantages and disadvantages of various experimental methods (e.g., accuracy, cost, applicable soil types, and measurement parameter ranges) to enhance analytical depth.

3、Section 6 presents a discussion on “Future Applications of Smart Sensors,” but the content remains rather general. It is recommended to expand this section by exploring:
(i) the potential of smart sensors for real-time monitoring and data fusion;
(ii) the integration of sensors with machine learning or inverse modeling techniques; and
(iii) issues of standardization and reproducibility in future research.

4、Section 5 again gathers a variety of experimental approaches but without sufficient cross-method comparison or evaluation. Including a summary table or matrix highlighting the performance metrics (accuracy, cost, soil applicability, and measurable parameter range) would substantially improve the review’s depth and readability.

Reviewer 2 Report

Comments and Suggestions for Authors

The manuscript entitled “Advances in Flow of Water Through Variably Saturated Soils: A Review of Numerical Models and Experimental Investigations With Use of Sensors” by Peter Osame, Ebikapaye J. Peretomode, and Haval Kukha Hawez addresses an interesting topic but requires major revision before it can be considered for publication in Sensors.

The paper presents a review on water flow through variably saturated soils, covering both numerical models and experimental techniques based on sensors. The topic is relevant within soil physics and hydrology; however, in its current form, the manuscript does not meet the standards of a scientific review article due to significant deficiencies in structure, literature update, critical analysis, and formal presentation.

Of the 132 references cited, only 12 (≈9%) correspond to the last six years. The review relies almost exclusively on classical works (Darcy, Richards, Van Genuchten), omitting recent advances in 3D modeling, inverse parameter estimation, IoT sensor networks, or artificial intelligence applications in soil hydrology. As a result, the manuscript reads more as a historical overview than as an up-to-date scientific review.

Section 4 (Models of fluid flow through…), although well organized into subcategories (single, dual-porosity, dual-permeability, and multiple-porosity models), is purely descriptive. Equation (3) contains a formal error (the term “–K” does not correspond to the standard formulation of Richards’ equation), and there is no discussion of the advantages or limitations of the different modeling approaches.

Section 6 (The future of smart sensors in soil column…) repeats verbatim ideas already presented in the introduction, such as the importance of studying water flow and the usefulness of sensors. It does not offer a genuine vision of the future: no emerging technologies, integration trends, or promising research directions are discussed. In its present state, this section should be completely rewritten to provide a forward-looking perspective rather than a repetitive summary.

The conclusion fails to fulfill its analytical role, merely reiterating previous content. It does not respond to the study’s objectives nor synthesize key findings or trends. A final reflection is needed on the most effective models, knowledge gaps, and potential directions for future research.

The figures lack uniformity and editorial consistency: they vary in font size, line thickness, typography, and graphical style. They appear to have been produced by different authors or sourced from heterogeneous materials without standardization, which hinders comparative interpretation.

The use of the Vancouver citation style is inconsistent throughout the manuscript. Frequent errors occur in the form of citation ranges ([2]-[5] instead of [2–5]), spacing and placement of citations within the text, as well as mismatches between in-text citations and the reference list. Moreover, the numerical order does not always follow the sequence of appearance. The problem is further compounded by a mixture of citation formats: alongside the numeric Vancouver style, some references appear in APA format, including author names and years in parentheses —for example, “(Simunek et al., 2003)” or “(de Rooij, 2000).” This inconsistency causes confusion and does not comply with the editorial requirements of Sensors, which mandates the uniform use of the Vancouver/MDPI referencing system. Therefore, the manuscript requires a thorough editorial revision to standardize all citations and references under a single, correct format.

I appreciate the authors’ effort in compiling a wide range of studies and encourage them to substantially revise and update the manuscript to strengthen its scientific contribution and alignment with the scope of Sensors.

Round 2

Reviewer 2 Report

Comments and Suggestions for Authors

The authors have submitted a revised version of their manuscript entitled “Advances in Flow of Water Through Variably Saturated Soils: A Review of Numerical Models and Experimental Investigations with Use of Sensors.” I appreciate their efforts to improve the paper and address the previous comments; however, after a detailed reading of both the revised manuscript and the response letter, the revision only partially meets the concerns previously raised and that the work still requires substantial improvement before it can be considered for publication. While some cosmetic adjustments have been made, the fundamental issues identified in the first review—namely the lack of analytical depth, the limited inclusion of recent literature, and inconsistencies in presentation—remain largely unresolved. I therefore recommend that the authors undertake a major revision, significantly updating the literature with post-2020 sources on IoT, AI, and 3D hydrological modeling, introducing critical comparisons between models, standardizing Figures and references, and rewriting the conclusion to provide a true synthesis of findings and future perspectives. Only after these major revisions would the manuscript reach the level of rigor and quality expected for publication in Sensors.

Although the authors mention that additional recent references have been incorporated and that Section 5 and the conclusion were rewritten, the literature update is minimal. Only a couple of new references have been added, both very general in nature, and the discussion continues to rely almost exclusively on classical works. The review still lacks coverage of major advances in the field, such as 3D modeling approaches, inverse parameter estimation, IoT-based sensing networks, or artificial intelligence applications for soil hydrology. Consequently, the paper continues to read more as a historical overview than as an up-to-date scientific review. This remains a major shortcoming.

In the section on numerical models, the explanation provided regarding the “–K” term in equation (3) is acceptable in principle, depending on the coordinate convention, but the text does not clarify this for the reader, nor does it correct or discuss the standard form of Richards’ equation. More importantly, this section remains entirely descriptive, offering no analytical or critical comparison of the different modeling approaches. This is a crucial aspect of a review article and is still missing in the revised version.

The section now titled “The Future of Smart Sensors in Soil Column Experiments” has been expanded to mention emerging technologies such as IoT systems, wireless sensor networks, UAVs, and AI-based monitoring. While this addition represents some improvement, the discussion remains very general and lacks supporting references, examples, or critical insights about integration, data management, or current research trends. It needs further expansion and connection with contemporary literature to truly reflect the state of the art.

The conclusion section has also been slightly modified, adding a brief reference to climate-smart agriculture and smart sensors. However, it still does not provide a synthesis of key findings, identification of knowledge gaps, or clear directions for future research. It continues to be read as a summary rather than a conclusive and analytical section.

Regarding presentation quality, the authors acknowledge that the figures were taken from different sources, but this issue remains unresolved. The figures still vary in style, font, and size, which breaks visual consistency. The referencing style has been improved but not fully corrected; some inconsistencies persist, with occasional mixtures of Vancouver and author–year formats that do not meet MDPI’s editorial standards.
